# Nrg1 is an injury-induced cardiomyocyte mitogen for the endogenous heart regeneration program in zebrafish

Matthew Gemberling, Ravi Karra, Amy L Dickson, Kenneth D Poss*

Department of Cell Biology, Howard Hughes Medical Institute, Duke University Medical Center, Durham, United States

**Abstract** Heart regeneration is limited in adult mammals but occurs naturally in adult zebrafish through the activation of cardiomyocyte division. Several components of the cardiac injury microenvironment have been identified, yet no factor on its own is known to stimulate overt myocardial hyperplasia in a mature, uninjured animal. In this study, we find evidence that Neuregulin1 (Nrg1), previously shown to have mitogenic effects on mammalian cardiomyocytes, is sharply induced in perivascular cells after injury to the adult zebrafish heart. Inhibition of Erbb2, an Nrg1 co-receptor, disrupts cardiomyocyte proliferation in response to injury, whereas myocardial Nrg1 overexpression enhances this proliferation. In uninjured zebrafish, the reactivation of Nrg1 expression induces cardiomyocyte dedifferentiation, overt muscle hyperplasia, epicardial activation, increased vascularization, and causes cardiomegaly through persistent addition of wall myocardium. Our findings identify Nrg1 as a potent, induced mitogen for the endogenous adult heart regeneration program.

*For correspondence: kenneth. poss@duke.edu

## Introduction

Myocardial infarction (MI) is a common injury that causes permanent loss of hundreds of millions of cardiac muscle cells, increasing susceptibility to heart failure and sudden death. Major goals of regenerative medicine are methodologies to enhance cardiomyocyte recovery after MI and to restore cardiac function to patients with heart failure. Although there has been much investment in candidate cardiac stem cell populations over the past decade (*Laflamme and Murry, 2011*; *Behfar et al., 2014*), many promising alternative strategies for heart regeneration have emerged, including activation of cardiomyocyte division, reprogramming of non-muscle cells into cardiomyocyte-like cells, and delivery of stem cell-derived cardiomyocytes (*Bersell et al., 2009*; *Qian et al., 2012*; *Shiba et al., 2012*; *Song et al., 2012*; *Chong et al., 2014*).

Heart regeneration occurs naturally after extreme tissue damage in non-mammalian vertebrates like zebrafish (*Poss et al., 2002*). New myocardium is created through division of spared cardiomyocytes, and lineage-tracing experiments have not yielded evidence for a stem cell contribution (*Jopling et al., 2010*; *Kikuchi et al., 2010*). Zebrafish regenerate after injuries that deplete 60% or more of their cardiomyocytes, suggesting broad potential of most or all cardiomyocytes to participate in regeneration in these animals (*Wang et al., 2011*). By contrast, cardiomyocyte division is robust through early postnatal life in mice and can enable regeneration, but by the adult stage is profoundly reduced (*Porrello et al., 2011*).

Factors that on their own can stimulate spontaneous creation of patterned, vascularized adult cardiac muscle would hold great potential for addressing human cardiovascular disease. The adult heart is famously resistant to forced hyperplasia, and tumors of myocardial origin are exceedingly rare. Adult mammalian cardiomyocyte proliferation has been reported to be stimulated to a minor

**eLife digest** Heart attacks—which are a major cause of death in humans—occur when a blocked blood vessel stops blood from flowing to the heart. This causes many heart muscle cells to die, which can result in permanent damage that makes survivors more susceptible to heart failure in the future. A major goal of regenerative medicine is to develop therapies that can improve the recovery of heart muscle cells after a heart attack and restore normal heart activity to patients with heart failure.

Unlike the human heart, the heart of an adult zebrafish is able to regenerate even after extensive damage. After an injury, the remaining heart muscle cells divide to replace the lost heart muscle, but it is not clear how this works.

A protein called Neuregulin1 (or Nrg1 for short) can stimulate heart muscle cells to divide. Gemberling et al. investigated the role of this protein in the regeneration of the heart in adult zebrafish. The experiments show that when the heart is injured, the gene encoding the Nrg1 protein is switched on in cells of the outer layer of the heart wall. When Nrg1 is deliberately activated in uninjured adult zebrafish hearts, it causes the muscle cells to divide, leading to many new layers of heart muscle forming over the course of several weeks. Along with promoting cell division, Nrg1 also makes the heart muscle cells return to an immature state more like stem cells.

Gemberling et al. found that Nrg1 also supports regeneration of the heart by changing the environment surrounding the muscle cells. For example, it stimulates the growth of new blood vessels and recruits non-muscle cells to the injury site. Gemberling et al.'s findings demonstrate that Nrg1 is sufficient to induce the growth of heart muscle growth in an adult animal, even in the absence of injury. To develop its therapeutic potential, future work will also need to identify how the gene that encodes Nrg1 is switched on by injury and identify the other molecules that interact with Nrg1.

extent after ischemic injury (*Senyo et al., 2013*). Genetic manipulations in cardiomyocytes of cell cycle genes like cyclin D, Rb, and/or p130 can increase cardiomyocyte cell cycle entry or division but do not cause significant muscularization (*Pasumarthi et al., 2005*; *Sdek et al., 2011*). Similarly, manipulations of Hippo signaling or miRNA function can increase cardiomyocyte proliferation after injury in mice (*Eulalio et al., 2012*; *Heallen et al., 2013*; *Xin et al., 2013*), but no compelling evidence for an impact on adult cardiogenesis has been reported. In adult zebrafish, several manipulations have boosted cardiomyocyte proliferation after trauma (*Jopling et al., 2012*; *Yin et al., 2012*; *Choi et al., 2013*), yet no genetic or pharmacologic method has stimulated cardiomyocyte proliferation or obvious cardiogenic growth in the absence of injury.

The extracellular factor Neuregulin1 (Nrg1) has multiple roles in cardiovascular biology and has been implicated as a cardiomyocyte mitogen. In developing zebrafish or mouse embryos, Nrg1 signaling is critical for cardiac myofiber trabeculation. Mice mutant in Nrg1, or in its receptors ErbB2 or ErbB4, die at mid-gestation, or later with cardiac-restricted mutations, from thin ventricular walls and aberrant trabeculation (*Gassmann et al., 1995*; *Lee et al., 1995*; *Meyer and Birchmeier, 1995*; *Liu et al., 2010*). Moreover, different forms of Nrg1 peptides have been delivered to adult animals, with various effects on the cardiovascular system. Nrg1 delivery to embryonic mouse cardiac explants or embryonic rat cardiomyocytes increased their proliferation in culture (*Zhao et al., 1998*), and Nrg1 treatment of cultured adult mouse cardiomyocytes or systemic injection into adult mice induced cardiomyocyte proliferation (*Bersell et al., 2009*). Although the proliferative increases in this latter study were relatively small, the authors also found evidence that Nrg1 injection improves cardiac repair after MI. Nrg1 has reported effects on cell survival, metabolism, angiogenesis, and myofiber structure in addition to cardiomyocyte proliferation (*Parodi and Kuhn, 2014*), influences that are possibly reflected by functional improvement after Nrg1 infusion in congestive heart failure patients (*Jabbour et al., 2011*).

It is unclear from these recombinant protein delivery experiments whether and how Nrg1 is part of an endogenous regeneration program. Moreover, corroborating evidence generated by other research groups for Nrg1 as a pro-regenerative cardiomyocyte mitogen is lacking. Indeed, a recent study argued that Nrg1 delivery to adult mice has no effects on cardiomyocyte proliferation (*Reuter et al., 2014*). To address this, we examined the role Nrg1 might play in the strong, endogenous regenerative response of the adult zebrafish heart. We find that zebrafish *nrg1* is upregulated in

perivascular cells following cardiac injury, and that blockade of Nrg1 signaling inhibits injury-induced cardiomyocyte proliferation. Most strikingly, transgenic reactivation of Nrg1 expression in the absence of cardiac injury stimulated many hallmarks of cardiac regeneration and markedly enhanced ventricular size. These findings implicate Nrg1 as a key mitogenic node between injury and the endogenous heart regeneration program.

## Results and discussion

### nrg1 expression is induced by cardiac injury

Using quantitative PCR, we found that *nrg1* levels increase after genetic ablation of ~50% of cardiomyocytes in the adult zebrafish ventricle. *nrg1* levels rise above baseline at 3 days post-injury and peak at ~11-fold above uninjured levels by 7 days, an injury timepoint at which cardiomyocyte proliferation also peaks (*Wang et al., 2011*). *nrg1* levels lower to ~fourfold above uninjured levels by 14 days post injury. To visualize *nrg1* expression, we used RNAScope, a modified in situ hybridization technique with improved sensitivity over standard methodology (*Wang et al., 2012*). While *nrg1* was rarely detectable in uninjured hearts, *nrg1* expression was induced 7 days after various injury methods. After resection of the ventricular apex, we observed *nrg1* staining in small regions surrounding cardiac damage. We saw larger stretches of expression after genetic cardiomyocyte ablation, distributed throughout the ventricular wall with occasional expression in the trabecular compartment (*Figure 1B–H*). *nrg1* signals in the ventricular wall were commonly in perivascular regions (*Figure 1D–F*).

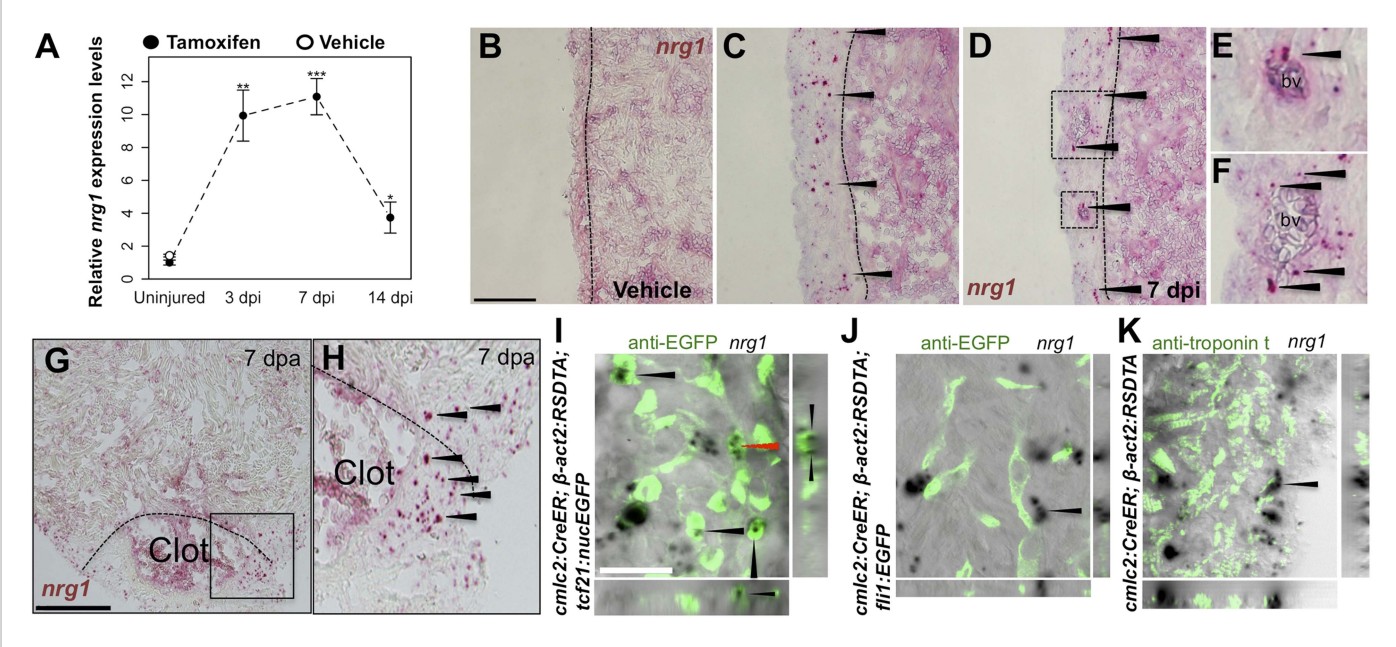

**Figure 1**. Induction of Nrg1 after cardiac injury. (**A**) Time course of *nrg1* induction in cardiac ventricles following severe genetic ablation of cardiomyocytes. *nrg1* mRNA levels were assessed by qPCR at 3, 7, and 14 days after ablation injury in tamoxifen-treated *cmlc2:CreER; β-act2:RSDTA* animals relative to control *cmlc2:CreER* animals (closed circles). *cmlc2:CreER; β-act2:RSDTA* (open circle) vehicle-treated animals serve as an additional control. Data are presented as mean ± SE. *p < 0.05, **p < 0.01, ***p < 0.001, Student's *t*-test, two-tailed. (**B–F**) Section images of in situ hybridization experiments assessed for *nrg1* expression in uninjured ventricles (**B**) or at 7 days after induced cardiomyocyte ablation (**C** and **D**). Dashed lines delineate the ventricular wall from the trabecular compartment. Higher magnification of boxes in (**E**) and (**F**) reveal *nrg1* signals surrounding vessels (bv). Arrowheads indicate examples of RNAScope signals. Scale bar represents 100 μm (**A** and **C**). (**G** and **H**) Section images of RNAScope in situ hybridization analysis for *nrg1* expression at 7 days after ventricular resection surgery. Image in (**H**) is a higher magnification of box in (**G**). Arrowheads indicate examples of RNAscope signals. Scale bar represents 100 μm. (**I–K**) Confocal slice images, with accompanying orthogonal views, of *nrg1* expression colocalized with *tcf21*:nucEGFP (**I**), *fli1*:EGFP (**J**), or cardiac muscle (Troponin, **K**). Arrows point to RNAScope signal, and red arrows indicate area for orthogonal views. Scale bar represents 20 μm.

To define the cells inducing *nrg1*, we combined RNAscope with transgenic reporter lines marking known cardiac cell types. In these experiments, the majority of *nrg1* signals could be localized to cells also positive for *tcf21*:EGFP fluorescence (*Figure 1I*). During heart development, *tcf21*⁺ cells contribute to the epicardial layer as well as vascular support cells (*Kikuchi et al., 2011a*; *Acharya et al., 2012*). After cardiac injury, epicardial cells and their progeny proliferate and incorporate into the injury site (*Kikuchi et al., 2011a*), where they have been assigned numerous pro-regenerative roles (*Gemberling et al., 2013*). In other contexts, Nrg1 is known to be expressed in Schwann cells and endothelial cells (*Meyer et al., 1997*; *Cote et al., 2005*; *Stassart et al., 2013*). Yet, we found that *nrg1* signals rarely overlapped during heart regeneration with cells positive for *fli1a*:EGFP or *cmlc2*: EGFP, which mark vascular endothelial cells and cardiomyocytes, respectively (*Figure 1J,K*). Thus, our results indicate that the dominant source of *nrg1* in the postnatal ventricular wall is the *tcf21*⁺ epicardial derived, perivascular cell compartment.

## Nrg1 signaling controls injury-induced cardiomyocyte proliferation

To test whether modulation of Nrg1 signaling can alter cardiomyocyte proliferation during regeneration, we employed loss- and gain-of-function approaches. Messages for Nrg1 receptors Erbb2 and Erbb4b were detectable by PCR methods in uninjured adult zebrafish ventricles (*Figure 2G*). Previous studies reported that the administration of AG1478, a small molecule inhibitor of Erbb receptors, mimics the effect of *erbb2* mutations on cardiac trabeculation in zebrafish (*Liu et al., 2010*). To examine Erbb activity requirements during regeneration, we treated adult zebrafish with 10 μM AG1478 from 6 to 7 dpa. We then quantified cardiomyocyte proliferation indices using nuclear markers of cardiomyocytes (Mef2) and cell cycle stage (PCNA), visual methodology that is required for accurate quantification of heart regeneration (*Wills et al., 2008*; *Wang et al., 2011*; *Yin et al., 2012*; *Fang et al., 2013*) (*Figure 2A,B*). This regimen decreased cardiomyocyte proliferation by ~54%, indicating that Nrg1 signaling is essential for heart regeneration (n = 20, 22; *Figure 2E*).

To increase Nrg1 levels, we created a transgenic line to inducibly express *nrg1* in cardiomyocytes when combined with a cardiomyocyte-restricted, taxmoxifen-inducible transgene, (*Tg(β-actin2:loxP-mTagBFP-STOP-loxP-Nrg1)*^pd107^; referred to hereafter as *β-act2:BSNrg1*) (*Figure 2F*; *Kikuchi et al., 2010*). We treated *cmlc2:CreER*; *β-act2:BSNrg1* animals with tamoxifen to induce *nrg1* expression (*Figure 2H,I*). 3 days later, we resected ventricles, and found that elevated *nrg1* expression led to an ~84% increase in the cardiomyocyte proliferation index near the injury site at 7 dpa (n = 15, 18; *Figure 2C–E*). This finding reveals a mitogenic influence of Nrg1 signaling on heart regeneration and suggests that *nrg1* levels are a limiting factor for cardiomyocyte proliferation after cardiac injury.

## Myocardial Nrg1 reactivation causes hyperplastic cardiomegaly

To test the effects of activating *nrg1* expression in the absence of injury, we treated 4- to 6-month-old *cmlc2:CreER*; *β-act2:BSNrg1* animals with tamoxifen and collected ventricles 7–30 days after treatment (dpt). Within 7 days of *nrg1* reactivation, there was a marked increase in cardiomyocyte proliferation, in particular within the ventricular wall (*Figure 3A,B*). Whereas the cardiomyocyte proliferation index in the trabecular compartment increased modestly from ~0.2% to ~1% after 7 days of *nrg1* overexpression (*Figure 3C*), the proliferation index of cortical muscle increased sharply from ~1.6% to ~28% (n = 8, 9; *Figure 3D,E*). Continued *nrg1* overexpression over an additional 7 days had similar effects, maintaining trabecular proliferation indices at ~1.6% and cortical muscle proliferation at ~32% (n = 10, 10; *Figure 3C–E*). This increased cardiomyocyte proliferation manifested through marked changes in cardiac anatomy. The ventricular wall thickened by ~76% at 7 dpt compared to controls, by ~265% at 14 dpt, and by ~459% after 30 days of overexpression (n = 8–11; *Figure 4A–E*). To test the extent to which cellular hypertrophy contributes to this wall thickening, we assessed the size and numbers of cardiomyocytes, which are predominantly mononuclear in adult zebrafish (*Wills et al., 2008*). We found large increases in the number of cardiomyocyte nuclei within the ventricular wall after 14 days of *nrg1* expression, and no evidence for increased cardiomyocyte size (*Figure 5A–C*). Thus, most or all of the effects of *nrg1* reactivation are hyperplastic. Such an extreme hyperplastic response to a single factor, which we term Nrg1-induced cardiac hyperplasia (iCH) for brevity, was unexpected from the published body of literature.

To determine long-term effects of iCH, we examined hearts 6 months after tamoxifen treatment. While animals displayed no outward differences from control clutchmates at this stage, analysis of dissected tissues revealed obvious cardiomegaly (*Figure 6A*). On average, experimental animals at 6

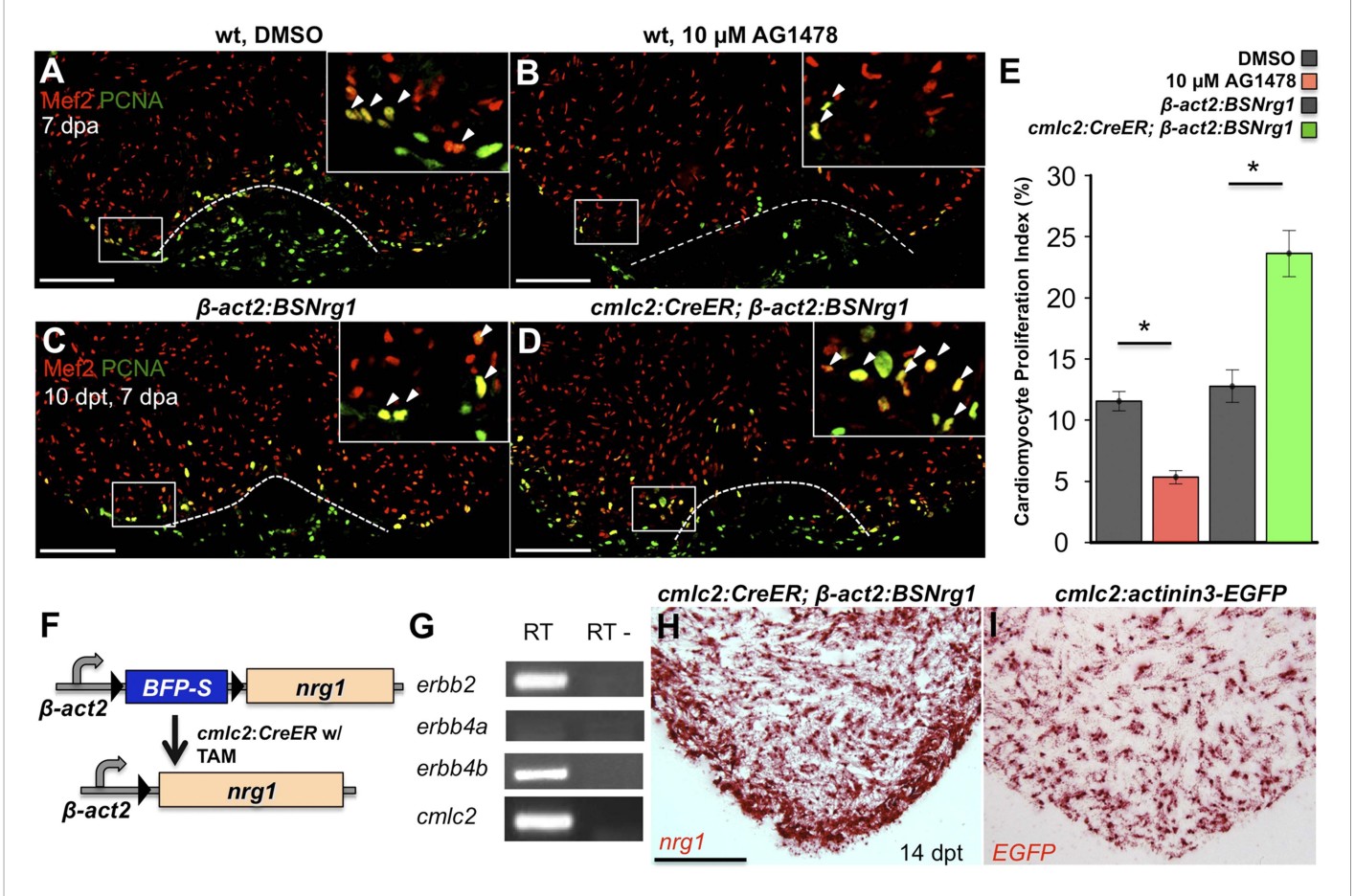

**Figure 2**. Nrg1 signaling modulates cardiomyocyte proliferation during regeneration. (**A** and **B**) Section images of injured ventricular apices of animals treated from 6 to 7 dpa with DMSO (**A**) or 10 μM AG1478 (**B**) and stained for Mef2+PCNA+ cells (arrowheads). Wounds are indicated by dotted lines. Scale bar represents 100 μm. (**C** and **D**) Section images of 7 dpa ventricular apices of control β-act2:BSNrg1 (**C**) or cmlc2:CreER; β-act2:BSNrg1 (**D**) animals treated with tamoxifen at 3 days before injury, stained for Mef2+PCNA+ cells (arrowheads). Scale bar represents 100 μm. (**E**) Quantification of cardiomyocyte proliferation at 7 dpa. DMSO-treated wild-type clutchmates (n = 22) were used as controls for 10 μM AG1478 treatment (n = 20), and tamoxifen-treated β-act2:BSNrg1 clutchmates (n = 15) were controls for cmlc2:CreER; β-act2:BSNrg1 (n = 18) animals. Data are represented as mean ± SEM. *p < 0.05, Mann–Whitney Ranked Sum Test. (**F**) Cartoon schematic of β-act2:BSNrg1 transgene. (**G**) RT-PCR results for erbb2, erbb4a, and erbb4b, indicating the presence of erbb2 and erbb4b messages in the uninjured adult ventricle. cmlc2 is shown as a control. (**H**) Section image of RNAscope in situ hybridization analysis for nrg1 expression at 14 days after tamoxifen-released expression in uninjured cmlc2:CreER; β-act2:BSNrg1 ventricles. (**I**) Section image of RNAscope in situ hybridization analysis for EGFP expression in uninjured cmlc2:actinin3-EGFP ventricles, used as a control to detect transgenic signals. Scale bar represents 100 μm.

months post-tamoxifen treatment had ventricles with a ~2.1-fold greater ventricular section area than control animals (n = 9, 10; *Figure 6B*). Tissue sections indicated elevated cardiomyocyte proliferation even at this late stage, although in some cases regions of the massively thickened ventricular wall showed mild fibrin and collagen deposition (*Figure 6C–G*). No fibrosis was detectable in hearts of animals after just 30 days of iCH (n = 7; *Figure 6H,I*).

To further understand the consequences of iCH, we monitored physiologic parameters over a period of 9 months. First, we measured cardiac function using echocardiography. The spatial resolution of standard B-mode echocardiography has limitations in zebrafish; yet we were able to use Doppler echocardiography to detect changes in atrioventricular inflow after 3 months of iCH (*Figure 7A,B*). These findings suggested that ventricular filling is altered as thickness of the ventricular walls increases (*Figure 7C*). Importantly, we were unable to detect cardiac dysfunction at 3 months of iCH, and instead our results suggested that stroke volume is enhanced with iCH (*Figure 7C,D*). By 3 months of iCH, ventricular wall hyperplasia was obvious by echocardiography (*Figure 7;*

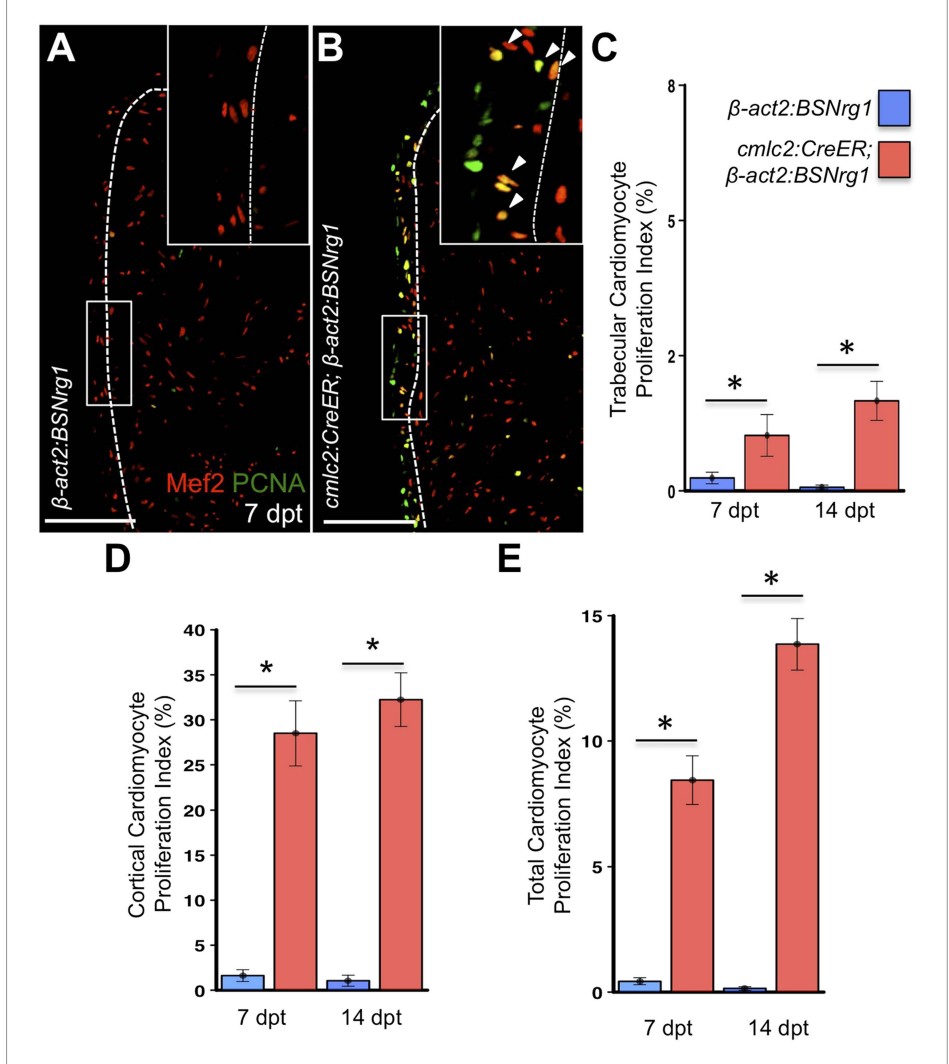

**Figure 3**. Nrg1 reactivation without injury induces proliferation of ventricular wall cardiomyocytes. (**A** and **B**) Section images from uninjured *cmlc2:CreER; β-act2:BSNrg1* and control ventricles at 7 days post-tamoxifen treatment (dpt), stained for Mef2⁺PCNA⁺ cells. Insets show high-zoom views of the boxed regions, and arrowheads indicate Mef2⁺ PCNA⁺ nuclei. Dashed lines delineate cortical (wall) from trabecular muscle. Scale bars represent 100 µm. (**C**) Quantification of cardiomyocyte proliferation in *cmlc2:CreER; β-act2:BSNrg1* and controls in the trabecular muscle compartment at 7 (n = 8, 9) and 14 dpt (n = 10, 10). Data are represented as mean ± SEM. *p < 0.05, Mann–Whitney Ranked Sum Test. (**D**) Quantification of cardiomyocyte proliferation in cortical muscle at 7 (n = 8, 9) and 14 dpt (n = 10, 10), from groups in (**A** and **B**). Data are represented as mean ± SEM. *p < 0.05, Mann–Whitney Ranked Sum Test. (**E**) Quantification of total cardiomyocyte proliferation at 7 (n = 8, 9) and 14 dpt (n = 10, 10), from groups in (**C** and **D**). Data are represented as mean ± SEM. *p < 0.05, Mann–Whitney Ranked Sum Test.

*Videos 1, 2*). Accurate echocardiography at 8 or 9 months of iCH was challenged by the extreme cardiac dysmorphology in these animals.

We next examined the ability of transgenic animals to swim against an increasing water current, an assay of cardiac function that is sensitive to massive cardiac damage and heart failure (*Wang et al., 2011*). Animals at 3 months of iCH showed normal endurance swimming, consistent with echocardiographic results. At later time points (8 or 9 months iCH), animals displayed a sharp reduction in swimming endurance, suggesting that continued addition of myocardium eventually turns deleterious (*Figure 7E*). Thus, forced *nrg1* reactivation in the absence of cardiac injury induces robust myocardial hyperplasia in adult zebrafish, adding many new layers of myocardium over weeks of continued cardiomyocyte proliferation.

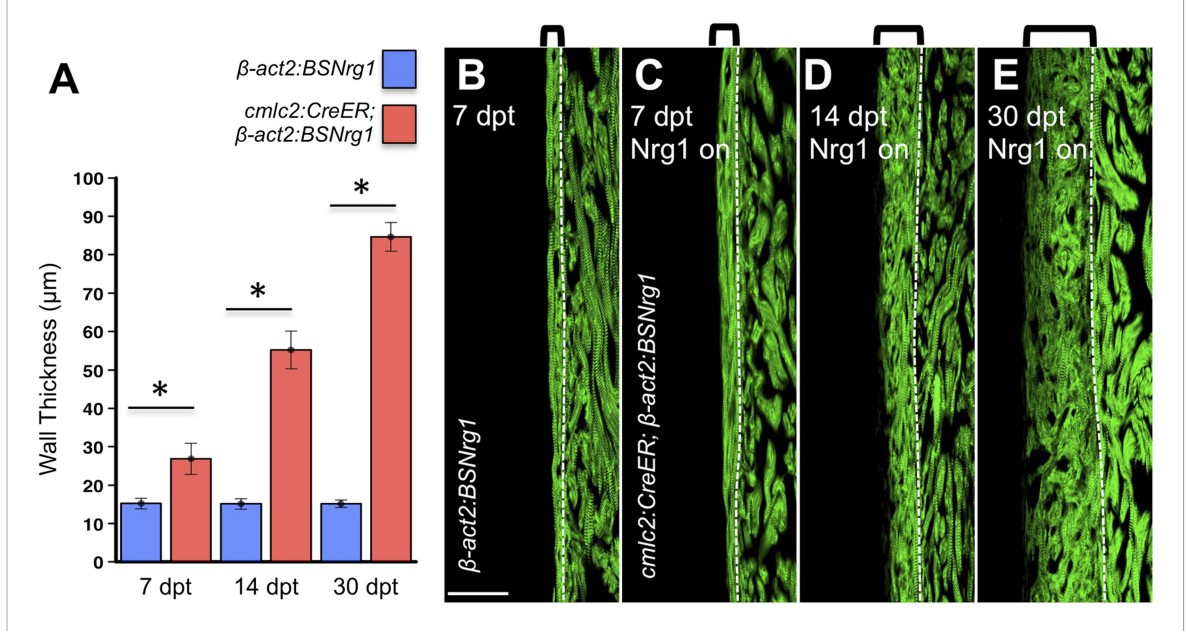

**Figure 4**. Nrg1-induced cardiomyocyte proliferation expands the ventricular wall. (**A**) Quantification of cortical muscle thickness at 7 (n = 8, 9), 14 (n = 10, 11), and 30 dpt (n = 11, 11). Data are represented as mean ± SEM. *p < 0.05, Student's *t*-test, two-tailed. (**B–E**) Section images of *cmlc2:CreER; β-act2: BSNrg1* (Nrg1 on) and control ventricles from 7 to 30 dpt, using animals also transgenic for *cmlc2:actinin3-EGFP* to indicate sarcomere organization. Brackets indicate cortical muscle, and dashed lines delineate cortical from trabecular muscle. Scale bar represents 100 μm.

## Myocardial Nrg1 reactivation induces the heart regeneration program

To identify mechanisms of iCH, we examined several hallmarks of injury-induced regeneration in zebrafish. First, to reveal the spatiotemporal growth patterns of cardiogenesis, we coupled iCH with multicolor clonal analysis. The cortical muscle in the ventricular wall typically forms from a small number of large cardiomyocyte clones. These clones expand laterally on the ventricular surface and

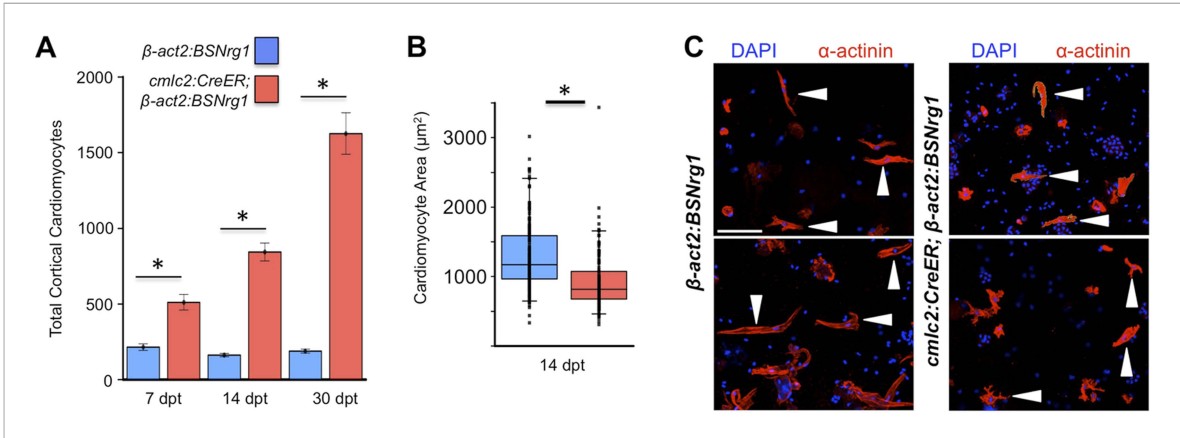

**Figure 5**. Nrg1 induces a hyperplastic, not hypertrophic, response. (**A**) Quantification of total ventricular wall cardiomyocytes in *cmlc2:CreER; β-act2: BSNrg1* animals and controls at 7 (n = 8, 9) and 14 dpt (n = 10, 10). Data are represented as mean ± SEM. *p < 0.05, Student's *t*-test, two-tailed. (**B**) Quantification of cardiomyocyte area in *cmlc2:CreER; β-act2:BSNrg1* animals and controls at 14 dpt. Data are represented as mean ± SD, with all data points represented. *p < 0.05, Student's *t*-test, two-tailed. (**C**) Confocal images of dissociated cardiomyocytes from *cmlc2:CreER; β-act2:BSNrg1* (right) and controls (left) at 14 dpt (n = 124, 172). Only cardiomyocytes with visible sarcomeres and nuclei were measured. Examples of quantified cells are marked with arrowheads. Scale bar represents 100 μm.

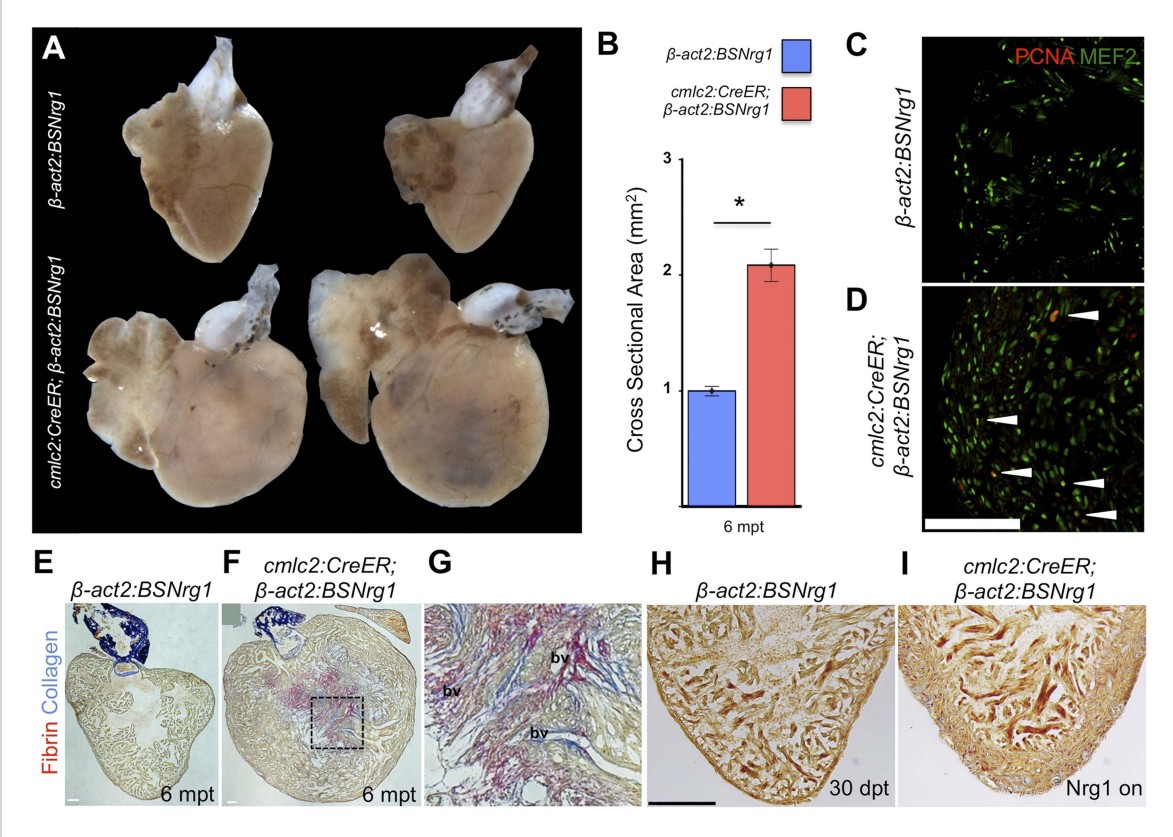

**Figure 6**. Nrg1-induced hyperplasia causes cardiomegaly. (**A**) Whole-mount images of *cmlc2:CreER; β-act2:BSNrg1* and control ventricles at 6 months post-tamoxifen treatment. (**B**) Quantification of the cross-sectional surface area of *cmlc2:CreER; β-act2:BSNrg1* (n = 9) and control ventricles (n = 10) 6 months post-treatment, revealing cardiomegaly effects of *nrg1* overexpression. Data are represented as mean ± SEM. *p < 0.05, Student's *t*-test, two-tailed. (**C** and **D**) Section images of ventricular walls of 6 mpt control *β-act2:BSNrg1* (**C**) or *cmlc2:CreER; β-act2:BSNrg1* animals (**D**) stained for Mef2[+] PCNA[+] cells (arrowheads). Scale bar represents 100 μm. (**E**) Section images of control *β-act2:BSNrg1* ventricles stained with Acid-Fuchsin Orange G (AFOG), revealing minimal collagen (blue), or fibrin deposition (red). Scale bar represents 100 μm. (**F** and **G**) Section images of *cmlc2:CreER; β-act2: BSNrg1* ventricles stained with AFOG, revealing collagen (blue) and fibrin deposition (red) in the inner portions of the thickened ventricular wall. Image in (**G**) is a high-zoom view of box in (**F**) and also indicates two examples of large coronary vessels (bv). (**H** and **I**) Acid-Fuchsin Orange (AFOG) staining reveals minimal fibrosis in *cmlc2:CreER; β-act2:BSNrg1* ventricle at 30 dpt despite the thickened ventricular wall (n = 7, 7). Scale bar represents 100 μm.

largely retain discernable boundaries through adulthood (*Gupta and Poss, 2012*). By contrast, regeneration of resected muscle occurs through roughly uniform proliferation by many cardiomyocytes near the injury site, generating a mixed conglomeration of small clones in the restored ventricular wall (*Gupta et al., 2013*). We induced *nrg1* expression simultaneously with permanent multicolor labeling in 5 weeks post-fertilization (wpf) juvenile animals, around the time of the initial emergence of cortical muscle. iCH caused ectopic wall thickening by 10 wpf, with obvious clone mixing and clone growth in the Z-plane away from the lumen (14/14 iCH, 0/11 controls; *Figure 8A–F*). These findings indicate that Nrg1 does not activate a gradual layering process but builds muscle radially with proliferation dynamics that are more reminiscent of injury-induced regeneration.

We next examined molecular signatures of iCH. After partial ventricular resection, cardiomyocytes near the injury activate regulatory sequences of the embryonic cardiogenic transcription factor *gata4*, before dividing to create new muscle (*Kikuchi et al., 2010*). Moreover, Gata4 activity is essential for regeneration of this muscle (*Gupta et al., 2013*). Using a transgenic reporter strain, we found that 7 days of iCH in mature adults induced *gata4:EGFP* expression in the outermost layer of cortical muscle (9/10 iCH, 1/11 controls; *Figure 8G,H*). Additionally, by coupling iCH with a transgenic reporter visualizing cardiomyocyte Actinin3 localization, we found that cortical muscle displayed reduced *cmlc2* expression and poorly organized sarcomeres (*Figure 8M–P*). These observations together suggest a reduction in the contractile program that is indicative of dedifferentiation. iCH also

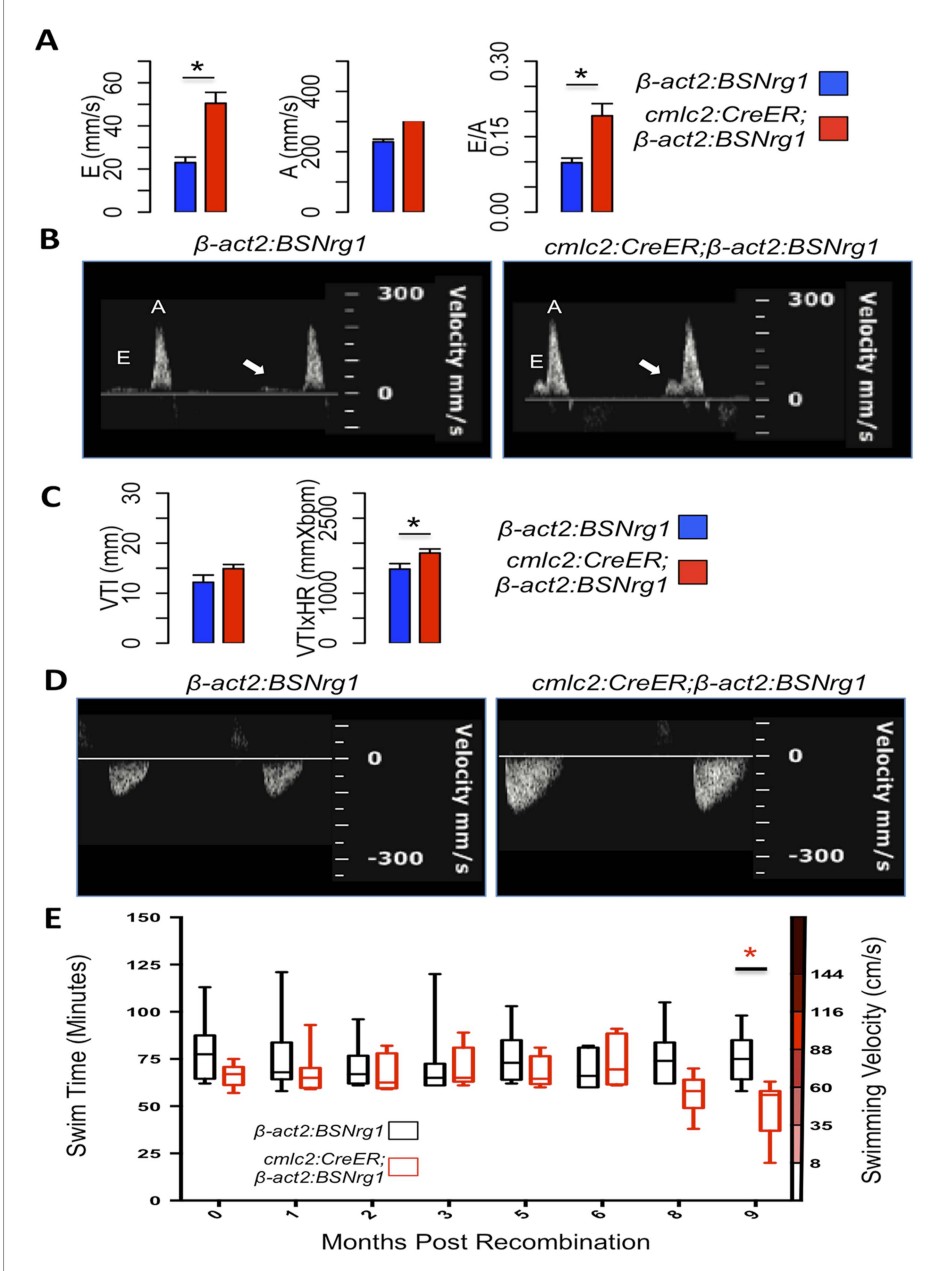

**Figure 7**. Effects of Nrg1 reactivation on cardiac function. (**A**) Doppler measures of ventricular filling obtained at the AV valve in *cmlc2:CreER; β-act2:BSNrg1* and *β-act2:BSNrg1* animals (n = 9 and 7). Data are represented as mean ± SEM. *p < 0.05, Student's *t*-test, two-tailed. (**B**). Representative PW Doppler at the AV valve in *cmlc2:CreER; β-act2: BSNrg1* and *β-act2:BSNrg1* animals. Arrows indicate E waves. (**C**) Doppler measures of cardiac output and stroke volume using the velocity time integral (VTI) obtained at the outflow tract (OFT) in *cmlc2:CreER; β-act2:BSNrg1* and *β-act2:BSNrg1* animals (n = 9, 7). Data are represented as mean ± SEM. *p < 0.05, Student's *t*-test, two-tailed. (**D**) Representative PW Doppler at the OFT in *cmlc2:CreER; β-act2:BSNrg1* and *β-act2:BSNrg1* animals. (**E**) Quantification of graded swimming performance of animals at varying times of *nrg1* overexpression plotted as box and whisker plots. Two-way ANOVA was performed looking at the effect of Nrg1 overexpression (p < 0.05), age (p = 0.38), and the interaction of Nrg1 overexpression and age (p < 0.05).

activated myocardial expression of *tgfβ3*, which was implicated previously in regeneration (4/5 iCH, 0/5 controls; *Figure 5I,J*) (*Chablais and Jazwinska, 2012*; *Choi et al., 2013*). We examined other cardiac cells types for their response to iCH. Retinoic acid synthesis in endocardial and epicardial cells

is induced by myocardial injury, where RA signaling is essential, but not sufficient, for cardiomyocyte proliferation (*Kikuchi et al., 2011b*). Fibronectin (Fn) synthesis is also activated in the epicardium by injury and is an essential extracellular matrix (ECM) component of regeneration (*Wang et al., 2013*). Both *raldh2* and *fn1* were induced in the epicardium by 7 days of iCH (7 of 9 iCH, 0 of 10 controls for each marker) consistent with the presence of a regenerative program (*Figure 5K,L,Q,R*). Along with epicardial marker induction, 7–14 days of iCH induced expansion of the epicardial layer that is reminiscent of that observed during regeneration (*Lepilina et al., 2006*; *Wang et al., 2011*) (*Figure 8S,T*). Finally, we assessed myocardial vascularization, which occurs concomitantly with injury-induced regeneration (*Lepilina et al., 2006*). The adult zebrafish ventricle typically has a thin muscular wall penetrated with vessels, but 30 days of iCH stimulated formation of major vascular network throughout the expanded ventricular wall (*Figure 8U,V*). After 6 months of iCH, many large coronary vessels were evident in the ventricle (*Figure 6G*). Together, this analysis indicates that expression of a single molecule, Nrg1, is sufficient to induce and maintain critical aspects of the heart regeneration program that encompass several cell types.

## Conclusions

Here, we identify the extracellular factor Nrg1 as a potent activator of the heart regeneration program in zebrafish. Nrg1 is induced by cardiac injury and pharmacological blockade of its receptor decreases injury-induced cardiomyocyte proliferation—each indicating involvement in the endogenous process. Moreover, whereas several experimental manipulations have been reported to increase cardiomyocyte proliferation in an injured adult heart, the Nrg1 protocol we describe here is potently cardiogenic in the absence of trauma.

Nrg1 was reported to have mitogenic effects on cultured adult cardiomyocytes (*Bersell et al., 2009*), and its most remarkable property in our experiments was the ability to induce and maintain adult cardiomyocyte proliferation in the absence of injury. Thus, the primary cellular target of Nrg1 is likely to be cardiomyocytes. Nrg1 effects additionally involve organizing a tissue microenvironment that includes expression of additional mitogenic factors, ECM regulation, and vascular perfusion. This recruitment of various non-myocyte cell types is likely to be an endogenous role of Nrg1, given that it is also known to stimulate vascularization after ischemic injury in the hindlimb and is capable of inducing expression of ECM components in fibroblasts (*Hedhli et al., 2012*; *Kim et al., 2012*). The next generation of genome editing-inspired tools for zebrafish researchers should enable tests of cell-restricted, inducible genetic deletion of key Nrg1 components in multiple cell types. These experiments promise to dissect and define Nrg1 signaling requirements for regenerative responses within the complex cardiac milieu.

Factors that on their own can stimulate the creation of full myocardial units have unique potential to avoid complications from immunosuppression and arrhythmia compared to cell-based approaches and can have straightforward pharmacologic entrypoints. Here, we have described effects of endogenous myocardial Nrg1 delivery, whereas previous studies have injected purified Nrg1

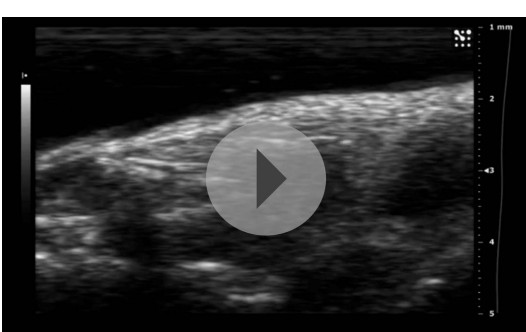

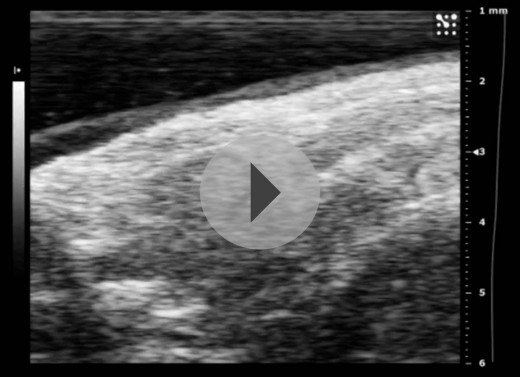

**Video 1.** B-mode video of echocardiography performed on adult control *β-act2:BSNrg1* animals 3 months post-tamoxifen treatment, indicating a thin ventricular wall. Relates to *Figure 7A–D*.

**Video 2.** B-mode video of echocardiography performed on adult *cmlc2:CreER; β-act2:BSNrg1* animals 3 months post-tamoxifen treatment, showing increased wall thickness. Relates to *Figure 7A–D*.

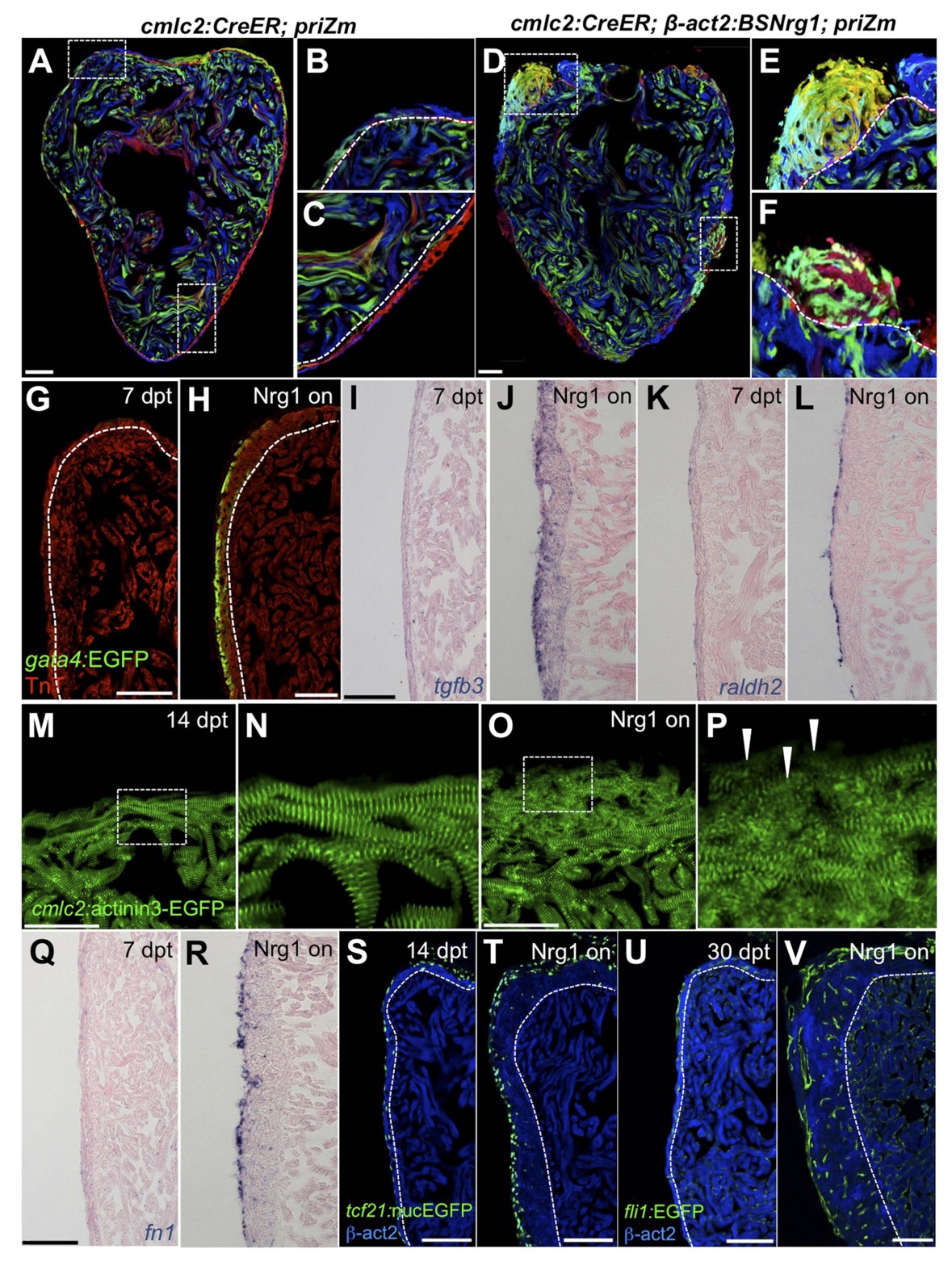

**Figure 8**. Nrg1 reactivation is sufficient to induce the heart regeneration program. (**A–F**) Section images of ventricles from control *cmlc2:CreER; priZM* (**A–C**) and *cmlc2:CreER; β-act2:BSNrg1; priZM* (**D–F**) animals treated with tamoxifen at 5 weeks post-fertilization (wpf) and assessed at 10 wpf. Cortical myocyte clones show clear boundaries between clones in control ventricles (**B** and **C**; n = 11). During *nrg1* overexpression, cortical muscle thickens appreciably via mixing and radial growth of distinct clones (**E** and **F**; n = 14). Dashed lines delineate cortical from trabecular muscle. Scale bar represents

*Figure 8. continued on next page*

**Figure 8. Continued**

100 μm. (**G** and **H**) Section images of ventricles from *cmlc2:CreER; β-act2:BSNrg1* (Nrg1 on) and control animals at 7 days post-treatment, using animals also transgenic for *gata4:EGFP*. EGFP induction is clear in the cortical layer during *nrg1* overexpression. Scale bar represents 100 μm. (**I** and **J**) Section images of ventricles from *cmlc2:CreER; β-act2:BSNrg1* and control animals at 7 days post-treatment, visualized for *tgfβ3* expression by in situ hybridization. Scale bar represents 100 μm. (**K**, **L**, **Q**, **R**) Section images of ventricles from *cmlc2:CreER; β-act2:BSNrg1* and control animals at 7 days post-treatment, visualized for *raldh2* (**K** and **L**) or *fn1* expression (**Q** and **R**) in epicardial cells by in situ hybridization. Scale bar represents 100 μm. (**M** and **N**) Section images of the ventricular wall from *cmlc2:CreER; β-act2:BSNrg1* and control animals at 14 days post-treatment, using animals transgenic for *cmlc2:actinin3-EGFP*. EGFP marks sarcomeric z-bands. Control animals (**M**) show organized sarcomeres in ventricular wall. *nrg1* overexpression (**O**) leads to reduced EGFP fluorescence and disorganization of sarcomeres. Arrowheads point to areas of reduced EGFP intensity and sarcomere organization. Boxes in (**M** and **O**) are represented as high-zoom in (**N** and **P**). Scale bars represents 50 μm. (**S** and **T**) Section images of ventricles from *cmlc2:CreER; β-act2:BSNrg1* and control animals at 14 days post-treatment, visualized for epicardial cells using a *tcf21:nucEGFP* transgene. *nrg1* overexpression grossly increases epicardial cell presence. Scale bar represents 100 μm. (**U** and **V**) Section images of ventricles from *cmlc2:CreER; β-act2: BSNrg1* and control animals at 30 days post-treatment, visualized for endothelial cells using a *fli1:EGFP* transgene. Increased endothelial cells and vasculature are evident in the thickened ventricular wall. Scale bar represents 100 μm.

protein (*Parodi and Kuhn, 2014*). These distinct experimental delivery methods could result in markedly different Nrg1 doses, target cells, and target receptor responses. It will be critical to define the mechanisms by which Nrg1 is induced by injury in zebrafish and restricted in the absence of trauma, as well as downstream Nrg1 targets in the regeneration program. Such investigation of the endogenous regulation of Nrg1 after cardiac injury can help guide methodology to optimize its delivery and impact on heart regeneration in mammals.

## Materials and methods

### Generation of *β-act2:BSNrg1* zebrafish

*nrg1* cDNA was amplified with the following primers and (Forward 5′-ACCGGTGCACCATGGCT GAGGTGAAAGCAGG-3′, Reverse 5′-GcGGCCGCTCACACAGCTATAGGATCCT-3′) and then subcloned into the AgeI/NotI site of the *β-act:loxP-TagBFP-STOP-lox-P* vector (*Gupta et al., 2013*). This construct was co-injected into one-cell-stage wild-type embryos with I-SceI. One founder was isolated and propagated. The full name of this transgenic line is *Tg(βactin2:loxP-mTagBFP-STOP-loxP-Neuregulin1)[pd107]*.

### Zebrafish

Wild-type or transgenic zebrafish of the hybrid EK/AB strain of the indicated ages were used for all experiments. All transgenic strains were analyzed as hemizygotes. Published transgenic strains or other alleles used in this study were *gata4:EGFP* (*Tg(gata4:EGFP)[ae1]*) (*Heicklen-Klein and Evans, 2004*); *cmlc2: CreER* (*Tg(cmlc2:CreER)[pd10]*) (*Kikuchi et al., 2010*) (used with *priZm*, *β-act2:BSNrg1*, and *bactin2:loxp-mCherry-STOP-loxp-DTA*); *tcf21:nucEGFP* (*Tg(tcf21:nucEGFP)[pd41]*); *bactin2:loxp-mCherry-STOP-loxp-DTA* (*Tg(bactin2:loxP-mCherry-STOP-loxP-DTA176)[pd36]*); and *cmlc2:actinin3-EGFP* (*Tg(myl7:actnb-EGFP)*) (*Wang et al., 2011*); *priZm* (*Tg(β-act2:Brainbow1.0L)[pd49]*) (*Gupta and Poss, 2012*); and *fli1:EGFP* (*Tg(fli1:EGFP)[y1]*) (*Lawson and Weinstein, 2002*). Ventricular resection surgeries were performed as described previously (*Poss et al., 2002*).

To induce expression of *nrg1*, adult *cmlc2:CreER; β-act2:BSNrg1* animals were bathed in 5 μM tamoxifen (Sigma-Aldrich, St. Louis, MO) for 18–24 hr, made from a 2 mM stock solution dissolved at room temperature in 100% propylene glycol. Animals were treated at a density of 3 per 125 ml of water and then returned to recirculating water. Juvenile *cmlc2:CreER; β-act2:BSNrg1; priZm* animals were incubated in 2 μM Tamoxifen from the same stock solution. Animals were treated at a density of 8 per 100 ml of water and then returned to recirculating water. For genetic cardiomyocyte ablation, adult *cmlc2:CreER; βactin2:loxp-mCherry-STOP-loxp-DTA* animals were placed in 0.3 μM Tamoxifen for 16 hr. Animals were treated at a density of 3–4 per 125 ml of water and then returned to recirculating water. Adult animals were incubated in 10 μM AG1478 (Selleck Chemical, Houston, TX) diluted from a 10 mM stock in DMSO for a 24-hr period from 6 dpa to 7 dpa.

### Quantitative PCR and RT-PCR

TriReagent was used to isolate RNA from whole ventricles, with 4–6 chambers pooled for each sample. From partially resected ventricles, only the apical halves were collected. A total of 0.5–1 μg of total

RNA was used in each cDNA synthesis reaction. cDNA was synthesized using the Roche Transcriptor first strand synthesis kit. Quantitative PCR was performed using the Roche Light Cycler 480, Roche UPL probes, and LightCycler 480 Probes Master. Intron spanning primer sets were designed using the Roche UPL design center. All experiments were performed using biological and technical triplicates. Primers were tested for efficiency and all primer sets were found to have efficiencies between 1.95 and 2.05. Primer sets used were ef1alpha (Forward 5′-CCTCTTTCTGTTACCTGGCAAA-3′, Reverse 5′-CTTTTCCTTTCCCATGATTGA-3′, used with probe #73) and nrg1 (Forward 5′-CACAAAT GAGTTCACATCACCA-3′, Reverse 5′-TCTGCTTTGCCATTACTCCA-3′, used with probe #76); *nrg1* levels were normalized to *ef1alpha* levels for all experiments.

RT-PCR was performed to assess *erbb* receptor levels. We used the following primers for each receptor: *erbb2* (Forward 5′-GATGGCAACATGGTTTTCCT-3′, Reverse 5′-TGGGTTCTCCACACTGTTCC-3′), *erbb4a* (Forward 5′-ATGTCAGGATCAGGGGATGA-3′, Reverse 5′-TTCCGATGGTTTACGAAAGG-3′), and *erbb4b* (Forward 5′-TTATTGCGGCAGGGGTTATTGGAGG-3′, Reverse 5′-CAACAGGAATCTT CACAGTCTCACCCTCA-3′).

## Histological analysis and imaging

In situ hybridization was performed on 10-μm sections of paraformaldehyde-fixed hearts as described (*Poss et al., 2002*). In situ hybridization was performed manually or with the aid of an InSituPro robot (Intavis). In situ hybridization data were imaged as described (*Kikuchi et al., 2011b*).

RNAscope (Advanced Cell Diagnostics, Hayward, CA) was performed on hearts fixed with paraformaldehyde for 1 hr at room temperature, equilibrated in 30% sucrose overnight, embedded in OCT, and sectioned to 10 μm. Tissue was washed with PBS for 5 min to remove OCT, followed by boiling in Pretreat 2 for 20 min. After Pretreat 2, slides were briefly washed with water and incubated for 10 min at 40°C with Pretreat 4. Following Pretreat 4, the manufacturer's protocol for RNAscope 2.0 HD detection Kit—Red was followed. Immunostaining was performed following *nrg1* detection as described in *Kikuchi et al. (2011b)*, with primary antibodies incubated overnight at 4°C. Advanced Cell Diagnostics designed *nrg1* and *EGFP* probes used in this study. RNAscope was performed on 6–8 animals for resection and ablation studies.

Primary and secondary antibody staining was performed as described (*Kikuchi et al., 2011b*). Acid Fuchsin-Orange G staining was performed on 10-μm sections as described (*Poss et al., 2002*). AFOG was performed on 6–10 animals per time point analyzed. Mef2/PCNA staining on sections from 7 dpa ventricles was performed and imaged as described (*Kikuchi et al., 2011b*). A Zeiss 700 confocal microscope was used to image Mef2/PCNA-stained sections from whole uninjured ventricles, using the tilescan function to acquire entire longitudinal sections from each ventricle. Images were taken of the three largest sections from each ventricle. Mef2+ and Mef2+/PCNA+ cells were counted manually. Three sections from each heart were averaged to compute a proliferative index for each animal. Cross-sectional areas of ventricles at 6-month post-tamoxifen or -vehicle treatment were calculated (ImageJ) using images of the three largest sections from each heart stained for TroponinT. *priZm* samples were imaged and processed as described (*Gupta and Poss, 2012*). A Zeiss 700 confocal microscope was used to image RNAscope for colocalization using a DIC filter to capture the *nrg1* signal. Z-stacks were taken for orthogonal views to show co-localization.

Primary antibodies used in this study: anti-PCNA (mouse; sigma) at 1:250, anti-Mef2 (rabbit; Santa Cruz Biotechnology, Dallas, TX) at 1:75, anti-troponinT (mouse; Thermo Scientific, Waltham, MA) at 1:100, anti-GFP (rabbit; Life Technologies, Carlsbad, CA) at 1:100, and anti-α-actinin (mouse; Sigma-Aldrich) at 1:400. Secondary antibodies used in this study: Alexa Fluor 594 goat anti-mouse IgG (H + L) for anti-Mef2, anti-PCNA, and anti-α-actinin; and Alexa Fluor 488 goat anti-rabbit IgG (H + L) for anti-Mef2, anti-PCNA, and anti-GFP. Secondary antibodies (Life Technologies) were all used at 1:200. Previously described probes for *raldh2*, *fn1*, and *tgfb3* were used for in situ hybridization (*Wang et al., 2011*; *Choi et al., 2013*). Raw data spreadsheets from experiments in this study are included in *Supplementary file 1*.

## Swimming endurance

Swim tunnel analysis was performed as described (*Wang et al., 2011*), with two exceptions: (1) fish were exercised in groups of 10–15 in a larger swim tunnel (Respirometer 5L [120 V/60 Hz]; Loligo cat #SW10060); and (2) there was no time limit for swimming. Swimming speed was increased every two

minutes after a 20 min acclimation period. Water velocities were measured up to 102 cm/s and values above were extrapolated using the other measures. Exhausted animals were removed from the chamber without disturbing the remaining fish, while others continued to swim. After all fish swam to exhaustion, they were allowed to recover and then placed back in recirculating water. To determine significance, 2-way ANOVA was performed looking at the effect of Nrg1 overexpression, time, and the interaction of Nrg1 overexpression and time. The number of total animals analyzed per time point: pre-recombination (24), 1 month (24), 2 months (24), 3 months (23), 5 months (21), 6 months (19), 8 months (17), and 9 months (17).

## Cardiomyocyte size measurements

Hearts were removed and washed in PBS wt/heparin and dissociated using a previously described protocol with minor changes (*Sander et al., 2013*). Three ventricles were placed in each tube and dissociated for 1.5 hr, leaving the majority of the ventricles intact while enriching dissociated cells for wall cardiomyocytes.

After dissociation, cells were plated onto a glass slide using a cytospin and spun for 3 min at 400×*g*. Then, cells were fixed for 10 min with 4% PFA and washed three times with PBS wt/0.1% Tween-20 for 5 min. Slides were stained for α-actinin overnight at 4°C and imaged using a Zeiss LSM 700 confocal microscope. Cells with the following criteria were measured using ImageJ software: (1) flattened appearance with visible sarcomeric staining; (2) clear dissociation from other cells; (3) single nucleus.

## Echocardiography

Echocardiography was performed on conscious zebrafish using a Vevo 2100 high-resolution imaging system with an MS-550S transducer (VisualSonics). Fish were sedated in phenoxyethanol and immobilized on a sponge immersed in fish water. Imaging was performed in the short axis (perpendicular to the fish) and long axis (parallel to the fish) using B-mode and PW imaging. Doppler parameters were measured using the Vevo2100 Cardiac Package by taking the average measurement of three consecutive contractions. Velocity time integral (VTI) and VTI*HR were used as measures of cardiac performance and output. Cardiac output could not be formally calculated, as outflow tract (OFT) diameter could not be directly measured by echocardiography. However, OFT diameter was not significantly different between experimental groups when measured in tissue sections (data not shown). Ventricular filling was measured by PW Doppler across the atrioventricular valve during diastole.

## Acknowledgements

We thank J Burris, N Lee, A Dunlap, and S Davies for zebrafish care, and Poss lab members for discussions and comments on the manuscript. We would like to thank H Rockman and L Mao for guidance with and use of echocardiography equipment. MG was supported by a predoctoral fellowship from the American Heart Association. RK was supported by a Clinical Investigator Award from NIH (K08-HL116485). This work was supported by a grant from NIH (R01-HL081674) to KDP.

## Additional information

### Funding

| Funder | Grant reference | Author |
| --- | --- | --- |
| National Institutes of Health (NIH) | R01-HL081674 | Kenneth D Poss |
| American Heart Association (AHA) | Predoctoral fellowship | Matthew Gemberling |
| National Institutes of Health (NIH) | K08-HL116485 | Ravi Karra |

The funders had no role in study design, data collection and interpretation, or the decision to submit the work for publication.

### Author contributions

MG, Conception and design, Acquisition of data, Analysis and interpretation of data, Drafting or revising the article; RK, Acquisition of data, Analysis and interpretation of data, Drafting or revising

the article; ALD, Acquisition of data, Analysis and interpretation of data; KDP, Conception and design, Analysis and interpretation of data, Drafting or revising the article

### Ethics

Animal experimentation: Work with zebrafish was performed according to an approved institutional animal care and use committee (IACUC) protocol (A100-12-04) at Duke University.

## Additional files

**Supplementary file**

• Supplementary file 1. Raw data spreadsheets used for quantification and statistical tests from experiments in this study.

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
