## [Decision Letter]

Thank you for sending your work entitled “Nrg1 is an injury-induced cardiomyocyte mitogen for the endogenous heart regeneration program in zebrafish” for consideration at *eLife*. Your article has been favorably evaluated by Janet Rossant (Senior editor) and 3 reviewers, one of whom is a member of our Board of Reviewing Editors.

The following individuals responsible for the peer review of your submission have agreed to reveal their identity: Marianne Bronner (Reviewing editor) and Benoit Bruneau (reviewer). A further reviewer remains anonymous.

The Reviewing editor and the other reviewers discussed their comments before we reached this decision, and the Reviewing editor has assembled the following comments to help you prepare a revised submission.

In this paper, Gemberling and colleagues examine the role of neuregulin signalling in regenerative cardiomyocyte proliferation in zebrafish. They show that neuregulin is upregulated after injury and that and pharmacological blockage reduces cardiomyocyte proliferation after heart resection whereas transgenically induced overexpression results in a higher proliferative index. Interestingly, overexpression of Nrg1 under normal conditions demonstrates its “sufficiency” to initiate expansion of the cardiomyocyte population in the absence of injury. This results in induction of cell behaviors and gene expression reminiscent of regenerative cardiomyocytes and a continuous expansion of the cortical heart muscle layer.

This is a nicely done study that convincingly shows a strong effect of reactivation of Nrg on cardiomyocyte proliferation in the normal heart. The results extend previous work suggesting that Nrg can promote some heart regeneration in mammals, albeit the results in rodents are relatively mild compared with present results in zebrafish. Additional characterization of the endogenous role of neuregulin and the effects of cardiomyocyte-specific overexpression would greatly strengthen the manuscript. With these additions, the manuscript would make a strong contribution.

1) Most of the work presented is based on over expression in cardiomyocytes. The sole loss of function analysis uses pharmacological blockade of the receptor, which is suggestive but not definitive. A genetic deletion approach to dissect the cell types in which Nrg1 is important for cardiac regeneration would be more appropriate.

2) It would greatly strengthen the manuscript to test whether specific activation of the neuregulin pathway in cardiomyocytes is sufficient for this dramatic effect. Only in this way would it be possible to distinguish whether the effect is cell-autonomous to cardiomyocytes or depends upon interactions with other cell types.

3) Figure 1 examines expression of Nrg7 days after injury and shows that it is upregulated at that timepoint. The authors should perform a more detailed time course, at least by qPCR, to look at the Nrg levels as a function of time.

---

## [Author Response]

*1) Most of the work presented is based on over expression in cardiomyocytes. The sole loss of function analysis uses pharmacological blockade of the receptor, which is suggestive but not definitive. A genetic deletion approach to dissect the cell types in which Nrg1 is important for cardiac regeneration would be more appropriate*.

While we agree a genetic loss-of-function for Nrg1 during regeneration would strengthen the manuscript, there are major obstacles that must be overcome. First, Nrg1 is required for embryonic development, which makes global Nrg1 null mutants unavailable for adult regeneration experiments. Thus a genetic loss-of-function experiment requires the creation of a conditional allele. As of now there are no reports in zebrafish of the successful generation of a conditional loss-of-function allele. It is important to introduce inducible gene deletion technology to highly regenerative systems like zebrafish, axolotl, and planarians, and we are actively working with others in the field on this. However, as this would take at least one year to complete and test, we are unable to provide the analysis here.

*2) It would greatly strengthen the manuscript to test whether specific activation of the neuregulin pathway in cardiomyocytes is sufficient for this dramatic effect. Only in this way would it be possible to distinguish whether the effect is cell-autonomous to cardiomyocytes or depends upon interactions with other cell types*.

We agree that cell type-specific activation or genetic loss of receptors would add to the understanding of the mechanism of Nrg1 function. Experiments to address cell-type specificity are actively being pursued in the laboratory and are an important next step, but are at least a year from completion due to the generation of numerous new transgenic lines to address these questions. Published reports show that Nrg1 can induce proliferation in cultured cardiomyocytes, suggesting that the effect is direct on cardiomyocytes. Current technology would allow us to express an activated form of the Erbb4 receptor specifically in cardiomyocytes, yet any downstream effect could be due to activation of Nrg1- or non-Nrg1-mediated pathways. Questions of cell type- specific activation or necessity in Nrg1-mediated enhancement of cardiomyocyte proliferation are important; yet as indicated above, attempts to acquire these secondary supportive data by the creation of novel tools would substantially delay our reporting the striking effects of Nrg1. This is current controversy over whether Nrg1 has mitogenic effects in the adult heart, amplified even more by a recent paper challenging this view published in December during the review of our manuscript ([35], PLoS One, 9(12): e115871). Therefore, we feel it is critical to report our primary findings as soon as possible without these new tools and supportive data.

To address points 1 and 2, we have added the following sentences in the revised manuscript, in the Conclusions:

*“*The next generation of genome editing-inspired tools for zebrafish researchers should enable tests of cell-restricted, inducible genetic deletion of key Nrg1 components in multiple cell types. These experiments promise to dissect and define Nrg1 signaling requirements for regenerative responses within the complex cardiac milieu.”

*3)*
Figure 1
*examines expression of Nrg7 days after injury and shows that it is upregulated at that timepoint. The authors should perform a more detailed time course, at least by qPCR, to look at the Nrg levels as a function of time*.

We agree that a time course would provide a more detailed understanding of Nrg1 mRNA dynamics following cardiac injury. To address this question, we have performed qPCR for Nrg1 mRNA levels at 3, 7, and 14 days following genetic ablation of cardiomyocytes. We find that Nrg1 mRNA levels are dynamic over time, with levels peaking at 7 days and then beginning to lower toward uninjured levels at 14 days post injury. Thus, peak Nrg1 levels correspond to the peak proliferative time point following genetic ablation, furthering support the finding that Nrg1 is an injury induced cardiac mitogen. We have included these results in a revised Figure 1, and described them in the first paragraph of the Results and Discussion section of the revised manuscript:

“*nrg*1 levels rise above baseline at 3 days post-injury and peak at ∼11-fold above uninjured levels by 7 days, an injury timepoint at which cardiomyocyte proliferation also peaks (45). *nrg*1 levels lower to ∼4-fold above uninjured levels by 14 days post injury.”